# Chemical Composition and In Vitro Ruminal Fermentation Characteristics of Native Grasses from the Floodplain Lowlands Ecosystem in the Colombian Orinoquia

**DOI:** 10.3390/ani13172760

**Published:** 2023-08-30

**Authors:** Mauricio Vélez-Terranova, Arcesio Salamanca-Carreño, Oscar M. Vargas-Corzo, Pere M. Parés-Casanova, José N. Arias-Landazábal

**Affiliations:** 1Facultad de Ciencias Agropecuarias, Universidad Nacional de Colombia, Palmira 763531, Colombia; 2Facultad de Medicina Veterinaria y Zootecnia, Universidad Cooperativa de Colombia, Villavicencio 500001, Colombia; 3Fedegan-Fondo Nacional del Ganado, Arauca 810001, Colombia; 4Institució Catalana d’Història Natural, 08001 Barcelona, Spain

**Keywords:** chemical composition, floodable savannas, in vitro fermentation, tropical forages

## Abstract

**Simple Summary:**

The floodplain ecosystem of the Colombian Orinoquia has a diversity of native grasses; however, there is little information on its agronomic and fermentation characteristics. The aim of this study was to determine the chemical composition and fermentation parameters of the grasses *Leersia hexandra*, *Acroceras zizanioides*, *Hymenachne amplexicaulis*, and *Urochloa arrecta* (Tanner grass, “introduced”) as a control group. Green forage samples were taken in a 1 m^2^ frame at 30, 40 and 50 days of age and biomass production was estimated. The chemical composition was analyzed by infrared spectroscopy, and the fermentation parameters by the in vitro gas production technique. Data were analyzed using a mixed model for repeated measures. The nutritional characteristics variations were: dry matter (DM, 0.7–2.0 ton/ha), crude protein (CP, 6.1–12.2%), neutral detergent fiber (NDF, 56.6–69.6%), ash (5.8–15.8%) and dry matter digestibility (DMD) between 20.8 and 60.6% from 12 to 48 h of fermentation. *L. hexandra* and *A. zizanioides* presented higher biomass, CP, ash, cellulose and Ca production. During the experimental period (30 to 50 days), the grasses did not present significant changes in nutrient availability. *L. hexandra* increased the concentrations of ammonia and total volatile fatty acids (VFA) and butyric acid. This last effect was also observed in the *A. zizanioides* grass. *L. hexandra* and *A. zizanioides* grasses constitute an alternative forage resource for ruminants during the flood seasons of the studied ecosystem.

**Abstract:**

Grasses from lowland ecosystems in flooded savannahs are useful to feed extensive grazing animals; however, scarce information about its agronomic and fermentation characteristics exists. This study aims to determine the chemical composition and fermentation parameters of native grasses from the floodplain lowlands ecosystem in the Colombian Orinoquia. Three native grasses (*Leersia hexandra*, *Acroceras zizanioides* and *Hymenachne amplexicaulis*) and a “control” grass (introduced *Urochloa arrecta*—Tanner grass) were sown and sampled at 30, 40 and 50 days of age. On each sampling date, biomass production in a 1 m^2^ frame was estimated, and the chemical composition and fermentation parameters were analyzed using near-infrared spectroscopy and the in vitro gas production technique, respectively. Data were analyzed using a mixed model for repeated measures and the least significant difference (LSD) was used for mean differentiation (*p* < 0.05). The grasses’ nutritional characteristics varied as follows: dry matter (DM, 0.7–2.0 ton/ha), crude protein (CP, 6.1–12.2%), neutral detergent fiber (NDF, 56.6–69.6%), ash (5.8–15.8%) and dry matter digestibility (DMD) between 20.8 and 60.6% from 12 to 48 h of fermentation. Native plants such as *L. hexandra* and *A. zizanioides* presented higher biomass production, CP, ash, cellulose, and Ca levels than the control plant. During the experimental period (30 to 50 days), the grasses did not present significant nutrient availability changes. In terms of fermentation characteristics, *L. hexandra* increased ammonia concentrations and total volatile fatty acids (TVFA) and butyric acid. This latter effect was also observed in *A. zizanioides* grass. *L. hexandra* and *A. zizanioides* grasses constitute a valuable alternative forage resource during the flooding times of the studied ecosystem.

## 1. Introduction

The Orinoquia region constitutes 25.4 million hectares of the Colombian territory of which more than 5 million consist of flooded savannahs located in the Arauca and Casanare departments. The flooded savannahs are an important ecosystem to support livestock production as one of the main economic lines in the region. The breeding and raising of extensive grazing cattle are the predominant livestock activities in the zone, where native grasses and legumes make up about 90% of the food available for animals [1,2]. The flooded savannahs consist of different physiographic units defined by the relief, water dynamics and drainage. Higher and lower zones are known as “banks” and “low”, respectively, and they are where native vegetation grows according to the hydrological dynamics determined by a monomodal rainfall regime, with rains from April to November and a dry period from December to March [1,3].

The low physiographic positions are zones that remain with a layer of water during the rainy season. The grasses and legumes within this ecosystem are adapted to the waterlogging conditions and frequently are affected during the dry period, where the layer of water disappears [4]. The soils are usually acid and of low fertility; however, native grasses such as *Acroceras zizanioides* (Kunth) Dandy (1931), (blackwater straw); *Hymenachne amplexicaulis* (Rudge) Nees (1829) (water straw); *Leersia hexandra* Sw. (1788) (Lambedora grass); and *Paratheria prostrata* Griseb. (1866) (Carretera grass), among others, can be found [1,5]. The grasses’ diversity of the “low” physiographic position is used to feed grazing animals under an extensive system and therefore with an unawareness about the nutritional and productive potential of these forage resources, just like the agronomic management and nutrient supply for animals. This has resulted in low productive and reproductive indices, reflected in low weight gains, and weaning weights (145–165 kg), high calving intervals (670–811 days) and low birth rates (33 to 45%) [1,2].

In order to take full advantage of the productive potential of a forage resource, it is required to know how its biomass production and chemical composition varies over time, as well as to determine the ruminal fermentation characteristics to identify the potential nutrients contribution to the animals and the production of other fermentation by-products of environmental importance (e.g., methanogenesis) [6]. According to the animals’ production level, forages, depending on their management and maturity, can provide most (if not all) of the nutrients required for an optimal performance in grazing livestock. Forages nitrogen, soluble and structural carbohydrates and minerals provide the energy, protein, minerals and vitamins required to maintain body functions and production traits, including respiration, body temperature, growth, reproduction and milk production [7].

Forage chemical composition analysis includes the estimation of dry matter, fiber fractions (cellulose, hemicellulose and lignin), crude protein (CP), energy content, ash and some specific minerals, among others [6], using traditional time-consuming laboratory procedures or through a faster and cheaper calibrated technique based on near-infrared spectroscopy (NIRS) [8,9]. To evaluate the rate and extent of digestion of the grasses’ chemical constituents, the gas production technique is one of the most used methods [10,11,12] that allows the estimation of several fermentation products such as volatile fatty acids (VFA), gases and microbial cell synthesis [13,14]. The chemical composition and fermentative characteristics information of the grasses will be useful for designing future in vivo assays [9,15] and to estimate animal performance in terms of milk, meat, reproduction, etc. [16]. Scarce information about the agronomic, nutritional and fermentative characteristics of native forages from the “low” physiographic position of the Orinoquia flooded savannahs is found. Most studies evaluated the nutritional and fermentative or only the fermentative characteristics of forage mixtures commonly observed under the “low” physiographic position [17,18]; however, there are few reports studying these characteristics in the native forages individually. For this reason, the aim of this study was to determine the chemical composition and fermentation characteristics of promising native grasses from the floodplain “low” ecosystem in the Colombian Orinoquia.

## 2. Materials and Methods

### 2.1. Study Site

This study was conducted at the Clarinetero Territorial Division Center, municipality of Arauca, eastern Colombia. The region is characterized by flat and floodable savannah topography with the presence of a “low” physiographic position (latitude: 7°08′17″ N, longitude: 70°59′59″ W, and altitude: 125 m) which constitute the lowest areas of the flooded savannah ecosystem (Figure 1), and present a diversity of pastures that serve as food for livestock, especially during the dry season, when forage availability is scarce. The soils have a sandy loam texture, and according to Holdridge’s classification, the region corresponds to a subhumid tropical forest zone [19].

During the research period (July–September 2021, rainy season), climate data were collected with a portable weather station located approximately 600 m from the experimental site. The highest and lowest average temperatures occurred in August (31.4 °C and 23 °C, respectively). The total precipitation during the grass growth cycle was 612 mm, and the average relative humidity was 91.8%

### 2.2. Evaluated Species

Native grasses included in the study were selected in a participatory manner with the livestock producers in the region. Previous meetings were held with livestock farmers associated with the livestock committee of the region to define the candidate grasses to evaluate according to their knowledge and observations of the most consumed forages by grazing cattle. Similarly, other criteria were based on aspects related to the species representativeness in the “low” physiographic position and the forage biomass production. *Urochloa arrecta* was included in the experiment as a “control” grass, as it is an introduced species commonly used in livestock systems in the region for grazing animals. The names and characteristics of the evaluated grasses (Table 1) were taken from Plants of the World Online.

### 2.3. Experimental Design

At the start of the rainy season in the area (May 2022), each species was triplicate established in the “low” physiographic position in 9 m^2^ plots (3 m × 3 m; 1 m distance between plots) under a completely randomized experimental design.

The agronomic management for the establishment of each experimental plot included manual soil preparation, sowing, irrigation and weed eradication. Approximately 500 g of diammonia phosphate (16% N and 40% P_2_O_5_) was added to each plot at planting to ensure adequate N and P availability for the plants [9,20]. Sowing was performed with viable vegetative material (stolon’s, rhizomes, bunch and stems) in morphological and sanitary terms. The vegetative material was obtained from farms near the experimental area. Ninety-three days after the plots were established, a leveling cut was made at 10 cm from the ground using a sickle, and the 50-day experimental period then began.

In each experimental plot, height measurements were taken from 15 random plants at 30, 40 and 50 days of age; the measurement was taken from ground level to the longest leaf using a metric rule. Then, three sampling points within each plot were randomly selected during plants emergence, and the cut-off ages (30, 40 and 50 days) were assigned to each measurement point randomized. These points were used to take grass samples, using a 1 m^2^ PVC frame by cutting the available material within the frame at 10 cm from the ground. On each sampling date, a grass sample was collected from each experimental plot. A Ranger precision balance was used to weigh the fresh sample obtained from each frame. Then, the samples were dried for 72 h at 60 °C in a Caloric brand electric oven. Based on the collected data, the fresh and dry matter yields per hectare were estimated. Each sample was stored in kraft paper bags and transported to the Analytical Chemistry Laboratory of the Agricultural Research Corporation (AGROSAVIA). The forage samples’ nutritional values were analyzed using near-infrared reflectance spectroscopy (NIRS [8]. The dry forage samples were homogenized to ensure a similar particle size; subsequently, they were placed in a 50 mm-diameter ring cup and spectra were obtained using FOSS NIR Systems DS6500 model equipment by scanning in the range of 400–2498 nm. The reference and the new spectra data were handled with WinISI 4.7.0.0 (Foss, Hilleroed, Denmark) [8]. The used calibration equations were constructed with spectra from 2020 forage resources of three families (Grass forage, *n* = 1418; legume forage, *n* = 320; and other forage plants, *n* = 282) sampled from 2014 to 2016 from different livestock regions of Colombia [8]. The estimated variables were the dry matter (DM), crude protein (CP), neutral detergent fiber (NDF), acid detergent fiber (ADF), lignin, hemicellulose (NDF—ADF), cellulose (ADF—lignin), ash (inorganic mineral fraction), calcium (Ca) and phosphorus (P).

### 2.4. In Vitro Ruminal Fermentation Analysis

Samples of the grasses (*A. zizanioides*, *U. arrecta*, *H. amplexicaulis* and *L. hexandra*) at 30 days of age were taken to submit them to short-term in vitro fermentation tests (48 h) following the methodology proposed [21]. Forage samples were ground and sieved through a 600 μM filter. Subsequently, 0.5 g of sample was disposed inside of 110 mL vials and mixed with 40 mL of culture medium [22] and 10 mL of ruminal inoculum preheated at 39 °C. The inoculum was obtained from three Holstein cows cannulated to the rumen, with a base diet of kikuyu grass (*Cenchrus clandestinus*) (Hochst. ex Chiov.) Morrone [23]. The inoculum of each animal was used individually and served as a repetition in statistical analysis. Fermentation flasks were prepared under constant CO_2_ pumping to maintain the anaerobiosis. The vials were sealed with rubber stoppers and aluminum clips and placed in a forced ventilation oven at 39 °C for 48 h.

Variables such as dry matter degradation (DMD), gas volume and pH were evaluated at 12, 24 and 48 h, while ammonia, methane, total volatile fatty acids (TVFA) and the acetic, propionic and butyric acids were studied at 24 and 48 h.

The DMD was determined using the gravimetric method. Gas volume was estimated using a pressure digital transducer (T443A—Bailey and Mackey, UK). The pressure data were used in a quadratic regression model estimated in the laboratory, to calculate the gas volume corrected for atmospheric pressure. The ammonia in ruminal fluid was determined using the potentiometric method with an ISE NH_3_ selective electrode. CH_4_ quantification was carried out by gas chromatography using Shimadzu GC-2014 equipment (Shimadzu, Kyoto, Japan) with an FID detector in a GC-Solution data station and manual injection with 25 µL VICI^®^ syringes. A 30 m long Stabilwax^®^—DA, 0.53 mm internal diameter and 0.25 µm film thickness was used. The standard used was 9.99% gaseous methane in balance with grade 5 nitrogen certified by Cryogas instrumental analysis. Nitrogen was used as the carrier gas with a flow set to 37.9 mL/min and an injector temperature of 150 °C. The methane production was estimated as a proportion of total gas volume. Finally, the VFA was also determined using gas chromatography with the same equipment and characteristics mentioned above. However, the standards used in this case were acetic, propionic and butyric acid, Fluka^®^ analytical grade. Nitrogen was used as the carrier gas with a flow set to 42 mL/min and an injector temperature of 250 °C. The fermentation process was carried out in the ruminal biotechnology laboratory (BIORUM) of the National University of Colombia—Medellín [23].

### 2.5. Statistical Analysis

As the evaluated agronomic, compositional and fermentation variables presented a longitudinal structure, data were analyzed using a mixed model for repeated measures [24] through the following linear model:

Y_ijk_ = μ + G_i_ + T_j_ + (GT)_ij_ + r_k_ + ε_ijk_
(1)

where Y_ijk_ represents the agronomic, compositional or fermentation variables observed in the “it” grass type and “jt” cutting age or fermentation time; μ represents the general average of the observed variable; G_i_ represents the fixed effect of “it” grass type (*L. hexandra*, *A. zizanioides*, *U. arrecta* (control) and *H. amplexicaulis*); T_j_ represents the fixed effect of “jt” cutting age (30, 40 or 50 days) or fermentation time (12, 24 or 48 h); (GT)_ij_ represents the interaction effect between grass type and cutting age or fermentation time; r_k_ represents the random effect corresponding to the kth repetition of each GT interaction; and ε_ijk_ represents the random error term. It was assumed that r_k_ and ε_ijk_ were independent and distributed ~N (0, s^2^). When it was required, the evaluated models were corrected for heteroskedasticity using functions that related residual variances to the mean. Three covariance structures were analyzed to consider the relationships between repeated measures (independent errors, compound symmetry and first-order autoregressive). The best model was selected using the Akaike and Bayesian information criteria. The analysis was performed using the “mixed and general linear models” option of the statistical InfoStat software [25]. The least significant difference (LSD) was used for mean differentiation (*p* < 0.05).

To evaluate the relationship between the grasses’ chemical composition variables and their fermentation parameters, multivariate profile diagrams were made through the InfoStat software [25].

## 3. Results

Most of the evaluated productive and chemical composition variables showed statistical differences throughout the studied experimental period (*p* < 0.05), except for the green forage yield (GFY), dry matter (DMY) yield, forage height, cellulose content and Ca levels. Mean values of grass type x cutting age interaction are shown in Table 2.

GFY and DMY presented differences between species and cutting ages (*p* < 0.05). GFY was the highest in *L. hexandra* and *A. zizanioides* (5.3 and 5.2 ton/GF/ha, respectively), followed by *H. amplexicaulis* and *U. arrecta* (control) (4.5 and 4.4 ton/GF/ha, respectively). A similar behavior was observed in terms of DMY, where the highest productivity was found in *L. hexandra* (1.9 ton/DMY/ha) followed by *A. zizanioides* (1.3 ton/DMY/ha). The species *U. arrecta* (control) and *H. amplexicaule* behaved similarly (1 ton/DMY/ha approximately in both cases). During the evaluated period, GFY and DMY increased obtaining the highest yields at 50 days (*p* < 0.05), with average values ranging between 4 and 5.4 ton/GFY/ha and 1.1 and 1.5 ton/DMY/ha.

Plant height varied between species and cutting ages (*p* < 0.05). *H. amplexicaulis* was the tallest plant (81.2 cm) followed by *U. arrecta* (control—60.0 cm), *L. hexandra* (56.1 cm) and *A. zizanioides* (40.4 cm). As expected, the lowest height was found at 30 days (47.7 cm) and the highest at 50 days (67.3 cm).

### 3.1. Nutritional Composition

During the entire experimental period, DM percentage was higher in the *L. hexandra* species (*p* < 0.05) with values between 33.5 and 39.1%. *A. zizanioides* was superior to the control plant at 30 days (27.8 vs. 19.8%, *p* < 0.05); however, this difference was not observed in the other evaluated ages. At 40 and 50 days, *A. zizanioides* and *H. amplexicaulis* were similar to the control (*U. arrecta*) with values between 22.5 and 26.2%.

Crude protein levels remained relatively stable in the native species *L. hexandra*, *A. zizanioides* and *H. amplexicaulis*, while in the control (*U. arrecta*) a decreasing behavior was observed. At 30 days, *L. hexandra* and *A. zizanioides* plants presented the highest CP levels (12.1 and 11.6%, respectively, *p* < 0.05) followed by *U. arrecta* (10.1%) and finally *H. amplexicaulis* (7.5%). At 40 days, *A. zizanioides* continued with the best performance (11.3%), while the rest of the plants were between 7.2 and 8.2%. On day 50, *A. zizanioides* and *L. hexandra* were superior to the rest of the plants with values of 12.2 and 10.6%, respectively.

*L. hexandra* showed the highest ash content during the entire experimental period (*p* < 0.05) with values between 13.8 and 15.8%. *A. zizanioides* and *H. amplexicaulis* were comparable to the control (*U. arrecta*) at 30 and 40 days (6.7–9.8%). By day 50, only *A. zizanioi-des* was superior to the control (10.9 vs. 5.8, respectively, *p* < 0.05).

Ca concentrations remained constant during the experimental period in *A. zizanioides* and *H. amplexicaulis*, while in *L. hexandra* and the control (*U. arrecta*) a decreasing response was evidenced. In general, *A. zizanioides* presented the highest Ca percentages during the study (*p* < 0.05; 0.49–0.59%). The other species were similar with values between 0.23 and 0.46%. Regarding P levels, *H. amplexicaulis* and *A. zizanioides* showed the highest values, with a statistical difference (*p* < 0.05) only at 40 days (0.25 and 0.32%, respectively). The control plant (*U. arrecta*) and *L. hexandra* presented the lowest values ranging between 0.14 and 0.23%.

The NDF varied among the species, with slight increases in *H. amplexicaulis* and the control (*U. arrecta*), while *L. hexandra* and *A. zizanioides* remained relatively constant. At 30 days, the NDF in the *A. zizanioides* and *L. hexandra* species was similar to the control (59.9 to 60.6%); however, it was lower at 40 and 50 days of growth (ranging from 56.6 to 61.6 vs. from 66.1 to 66.9%, in the control plant). The *H. amplexicaule* plant presented the highest NDF values during the experimental period (66.3 to 69.6%). The control plant (*U. arrecta*) presented the lowest ADF content at 30 days (29.8 vs. 37.8–40.6%). No differences were observed between the native species and the control plant at 40 days of age, with values between 35.4 and 37.1%. Finally, at 50 days, both the control (*U. arrecta*) and *L. hexandra* attained the lowest ADF values (34.6 and 37.2, respectively).

At 30 days of age, the lignin content was the lowest in the control plant (6.5 vs. 9.5–10.4%; *p* < 0.05); although, at 40 and 50 days no differences were observed among the studied species with values ranging between 8.1 and 9.6%.

The hemicellulose content at 30 days was superior in the control plant compared to the native grasses (30.5 vs. 23.3 on average for native plants). At 40 and 50 days, the control and *H. amplexicaulis* had similar content while this was higher than that obtained in *A. zizanioides* and *L. hexandra*, with values ranging from 30.5 to 32.3% (*p* < 0.05). The cellulose concentration only varied between species (*p* < 0.05) and was highest in native species (28.6% on average) compared to the control plant (24.9%).

Although in the evaluated plants, on average, the maximum forage yield (GFY and DMY) was reached at 50 days of age, their best performance in terms of DM, CP, NDF, ADF, lignin and P nutritional composition variables was at 30 days of age, without any drastic changes until day 50. Even some variables such as ash and Ca content attained their highest values at 30 days.

### 3.2. Grasses Fermentation Characteristics

The fermentation parameters of the grasses varied according to the species, measurement time and the species x measurement time interaction. Variables such as pH, acetic and propionic acids were influenced only by the measurement time (*p* < 0.05). The pH was higher at 12 h compared with the other measurement times (6.7 vs. 6.6), while acetic and propionic acids were superior at 48 than 24 h (36.4 vs. 15.1 and 16.8 vs. 8.1, respectively).

Gas volume, TVFA and butyric acid were affected by the grass species and measurement time independently (*p* < 0.05). *H. amplexicaulis* presented the lowest gas volume compared with the other plants (*p* < 0.05; 152.7 vs. 1606 mL/g DM on average); similarly, the gas volume increased during the evaluated measurement times with values of 124.3, 155.0 and 196.7 mL/ g DM for 12, 24 and 48 h, respectively (*p* < 0.05). TVFA was higher in *L. hexandra* compared with the other plants (*p* < 0.05; 45.5 vs. 41.8 mmol/L on average). Additionally, higher TVFA levels were found at 48 than 24 h (*p* < 0.05; 60.3 vs. 25.2, respectively). A similar result was observed with butyric acid concentration. In this case, *L. hexandra* presented the highest value (*p* < 0.05, 5 mmol/L), followed by *A. zizanioides* (4.7 mmol/L), while *U. arrecta* (control) and *H. amplexicaulis* were similar (4.3 mmol/L on average). Butyric acid was superior (*p* < 0.05) at 48 h (7.1 mmol/L) respect to 24 h (2.1 mmol/L).

Finally, DMD, ammonia and methane variables varied by the species x measurement time interaction. Fermentation parameters of the interaction are shown in Table 3 and Table 4. *U. arrecta* (control) presented a superior DMD compared with the native species in all of the evaluated measurement times (*p* < 0.05), with values of 31.9, 48.8 and 60.6% at 12, 24 and 48 h, respectively. Native grasses showed DMD mean values of 24.4, 37.1 and 49.5% at 12, 24 and 48 h, respectively. The ammonia was higher at 24 and 48 h in the *U. arrecta* (control) and *L. hexandra* species (*p* < 0.05) than the other grasses, with values ranging from 42.2 to 52.6 mg/dL. Methane at 24 and 48 h of fermentation was similar between the studied plants. The only difference observed was at 12 h of fermentation was that *H. amplexicaulis* presented higher methane concentrations than the *L. hexandra* species (*p* < 0.05; 33.8 vs. 27.8 mL/g DMD).

## 4. Discussion

Forage yield varied among the studied grasses with value ranging from 4.4 to 5.3 ton/GFY/ha and 1 to 1.9 ton/DMY/ha. The GFY and DMY values are within the range observed in tropical pastures adapted to the floodplain “banks” ecosystem in the Colombian Orinoquia, and other grasses cultivated in the dry season with 6 to 8 weeks of age [9,26,27]. The *U. arrecta* (control) plant showed an average DMY of 1.0 ton/ha which is a lower value compared with the 1.3 to 1.5 ton/ha found in another study without any type of fertilization [28].

Under the lowland conditions of the studied ecosystem, native grasses such as *L. hexandra* and *A. zizanioides* behaved better than the control in terms of biomass production, with values ranging among 5.2 to 5.3 and 1.3 to 1.9 ton/ha for GFY and DMY, respectively. These yield differences could be attributed to several causes. Although the control plant (*U. arrecta*), *L. hexandra* and *A. zizanioides* present a similar growth habit (Stoloniferus), differences in grass height were observed, the control grass being one of the tallest, followed by *L. hexandra* and *A. zizanioides*. Likewise, in the case of native grasses a greater forage biomass density was also observed. Thus, the results suggest that native grasses’ height together with a better forage density as a consequence of a proper adaptation to the study area, allowed the observed biomass differences to be explained. Similarly, even though the *U. arrecta* (control) plant is recognized for growing in swampy and flooded grasslands [29], it seems that prolonged flooded periods such as those of the study area (6 months approximately) could adversely affect its establishment and therefore the forage yield.

Grasses showed the greatest GFY and DMY at 50 days of age with values between 4 and 5.4 and 1.1 and 1.5 ton/ha, respectively. These yields are lower than the 4.4 to 8.5 ton/GFY/ha and 1.5 to 2.5 ton/DMY/ha observed in native grasses of “banks” physiography positions present in the same studied ecosystem [9]. The lower forage yields obtained in the lowland grasses could be attributed to the oxygen deficiencies arising in flooded conditions, limiting plants root respiration, and causing delayed or stunted growth [30]; thus, a reduced biomass production is expected in grasses under waterlogging conditions.

### 4.1. Grasses Chemical Composition

In terms of DM, CP, cellulose, ash, Ca and P content, native grasses performed better or similar than the control plant *U. arrecta*. *L. hexandra* was remarkable in the DM (33.5 to 39.1%) content compared with the other species. Although waterlogged stressed plants are known to reduce the dry matter production [31], it seems that the adaptive mechanisms of *L. hexandra* allow it to withstand the adverse flood conditions. Since DM in *L. hexandra* is high from an early to medium age (30 days onwards), it is expected that the cell content fraction of the DM (sugar, starches, protein, fat, pectins, etc.) predominate over the fiber fraction for a longer period, allowing the pasture digestibility to last longer over time which is useful in livestock systems with long grazing periods [9,32].

The native grasses’ CP content was not significantly affected by the 50-day experimental period, while in the control grass a decreasing behavior was observed. In general, the grasses’ CP concentration is expected to reduce over time as the plants’ metabolic activity also decreases through the age [32,33]. The relative constant CP concentration during the evaluated experimental period in the native grasses suggest that CP reduction may be more noticeable after 50 days of age. This indicates that these grasses reach maturity at slower rates, making an efficient use of the available nitrogen in different metabolic processes. Similar results were found in the *A. compresus* and *P. plicatulum* native grasses from the floodplain “banks” ecosystem [9]. Likewise, it has been reported that in the “low” soils of the evaluated ecosystem, a greater mineral nutrients availability (including nitrogen) can be found compared to other physiographic positions such as the savannah “banks” (0.78 vs. 0.66 g/Kg of total nitrogen in the “low” and “banks” physiographic positions, respectively) [4]. The greater nutrient availability, in conjunction with the waterlogging, might facilitate that different forms of nitrogen and other elements enter the soil solution and remain disposed to be assimilated by plants root for their metabolism and production [34]. In terms of CP content, native grasses such as *A. zizanioides* and *L. hexandra* were superior to the control plant during most of the studied growth phases (values ranged between 8.1 and 12.2%), which makes them an important protein source for the livestock activity in the study area.

The mineral content of some native grasses outperformed the control grass. The ash and Ca levels were the highest in *L. hexandra* and *A. zizanioides* plants (from 13.8 to 15.8% and from 0.49 to 0.59% for ash and Ca levels, respectively), during the studied 50-day experimental period, while the P content was remarkable in *H. amplexicaule* and *A. zizanioides*, especially at 40 days of age (0.25 and 0.32%, respectively). The ash content in *L. hexandra* was superior to the values observed in several *Brachiaria* accessions [35], native grasses of “banks” ecosystems [9] and tropical grasses from the Brazilian semiarid regions [33]. This result suggests that *L. hexandra* presents a high nutrient extraction capacity. This property allows it to use the greatest nutrients availability reported in the “low” physiographic position [4] to ensure a higher DM and ash concentration. Some research reported that the *L. hexandra* extraction capacity has been used to decontaminate soils with high levels of chromium or weathered oil and to bioestimulate the microbial activity in degraded soils [36,37]. The mineral composition found in *L. hexandra* constitutes a useful resource to feed grazing animals; however, it is important to identify and quantify the mineral component of the ash fraction to ensure safe use in animal nutrition.

The Ca and P content observed in *A. zizanioides* and *H. amplexicaulis* are similar to other forages under tropical environments, with values within the optimal range for cattle feeding [38].

The grasses’ NDF ranged from 56.6 to 69.6%, consistent with the values reported in grasses grown in the tropical regions of Mexico and Brazil [34,38]. In grasses such as *H. amplexicaulis* and the control grass (*U. arrecta*), the NDF tended to increase over the experimental period. This is an expected behavior since plant cell wall components (cellulose, hemicellulose and lignin) increase as grasses reach maturity [33]. In the case of *L. hexandra* and *A. zizanioides*, the NDF levels remained stable, ranging from 56.6 to 61.6%, attaining statistically lower values at 40 and 50 days of age compared to the control plant. This result suggests that in *L. hexandra* and *A. zizanioides*, the fiber components accumulation occurred at slower rates during the evaluated experimental period. A similar result was obtained with the *Axonopus compresus* grass under the “bank” physiographic position from the floodable savannah [9].

In the NDF fraction, the control plant showed a higher hemicellulose concentration among the studied species; however, the native grasses presented higher cellulose levels, which is advantageous since cellulose is the main polysaccharide of the primary and secondary cell wall that is potentially degraded by rumen microbes (bacteria, protozoa and fungi) to produce glucose. In the case of hemicellulose, despite the fact that it is also a potentially degradable polysaccharide, its degradation is less since it is bound to the indigestible lignin fraction which results in a lower digestibility [39,40].

In terms of ADF and lignin content, the values observed in the grasses ranged between 29.8 and 40.6% and 6.5 and 10.4%, respectively, comparable to what was found in other native grasses of the evaluated ecosystem, *Brachiaria brizantha* (A. Rich.) Stapf, *Brachiaria humidicola* (Rendle) Schweick and *Paspalum notatum* Flüggé under tropical conditions [9,38]. Despite the control grass presenting a lower ADF and lignin content at 30 days of age, no differences in respect of native grasses were found at other measurement times.

Although grasses on average attained their maximum forage yield (GFY and DMY) at 50 days of age, their chemical composition (DM, CP, NDF, ADF, lignin, ash, P and Ca) was stabilized from 30 days of age without drastic changes. This trend is concordant with another report in tropical grasses where the maximum biomass and DMY were reached at 63 days, while the maximum CP and cellular content were observed at 21 days of age [32]. This result suggests that under lowland conditions of the studied ecosystem, the evaluated plants could be harvested during the studied period to feed animals without greatly affecting the macro and micronutrients availability. Similarly, a greater quality loss is expected after 50 days of age. This is consistent with the result observed in two *Urochloa* lines (Basilik and Cayman) and *Megathyrsus maximum* cv. Mombaza, where the optimal grazing point in tropical conditions was between 4 and 6 weeks of age, with greater quality loss after 8 weeks [41]. It has also been reported that grasses such as *Andropogon gayanus* Kunth, cv. Planaltina or Massai grass (*Panicum maximum* × *Panicum infestum* cv. Massai) can be managed without losses between 21 and 63 days of age under a Brazilian semi-arid region [42].

### 4.2. In Vitro Fermentation Characteristics

Most of the grasses’ fermentation parameters did not change throughout the experimental period. Despite the pH varying according to the measurement time, its value ranged from 6.6 to 6.7 which is within the acceptable range to guarantee the optimal degradation of organic and fibrous components of feeds in in vitro fermentation systems [43].

The control grass showed the highest DMD among the grasses with percentages between 31.9 and 60.6% from 12 to 48 h of fermentation. These results are comparable with the DMD observed in *Urochloa hybrid* cv. Cayman, *Megathyrsus maximus* cv. Mombaza, *Urochloa brizantha* cv. Toledo and *Urochloa decumbens* grasses, fermented at ages between 42 and 65 days [44,45]. Although the control grass showed the highest DMD, a larger gas production was expected given the positive correlation between these two variables [46]; however, no differences were observed in respect to *L. hexandra* and *A. zizanioides*. Even these two grasses presented a superior butyric acid concentration than the control grass, and in the case of *L. hexandra* a higher TVFA level was also observed.

In the gas production technique, several fermentation products are formed (VFA, CO_2_, CH_4_, microbial cells, etc.); however, the gas volume produced is mainly related to the VFA formation [10,47]. In this sense, the fermentation characteristics of the *L. hexandra* native grass is more advantageous than the control grass, since the TVFA production is encouraged with a higher proportion of butyric acid. The latter effect was also observed in the *A. zizanioides* native plant. TVFAs (acetate, propionate and butyrate) are important because they constitute the main energy source for the animal. The butyric acid is the principal energy source of epithelial cells in ruminants and promote cell proliferation that leads to improved feed degradation and utilization [48,49]. In this way, greater production efficiency of energy sources for the animal is guaranteed with the use of *L. hexandra* and *A. zizanioides* grasses.

The highest ammonia production was found in the control and *L. hexandra* species at 24 (42.8 and 42.2 mg/dL, respectively) and 48 h (51.6 and 52.6 mg/dL, respectively) of evaluation. These results are similar or even higher than the values obtained in temperate (*Cenchrus clandestinus* (Hochst. ex Chiov.) Morrone or tropical (*Megathyrsus maximus* (Jacq.) B.K.Simon and S.W.L.Jacobs, *Urochloa humidicola* (Rendle) Morrone and Zuloaga) grasses, fermented during 24 and 48 h at ages from 30 to 42 days [50,51]. The values of ammonia levels observed in the control and *L. hexandra grasses* are a consequence of the fact that these grasses presented the highest CP levels at 30 days (10.1 and 12.1%, respectively). CP is degraded to form ammonia, which is used as a source of nitrogen by ruminal microorganisms and is therefore a determining factor in the synthesis of microbial cells [52]. The ammonia contribution provided by these grasses is above the 200 mg/L recommended for an adequate nitrogen supply in diets based on low digestibility tropical forages [53]. These results indicate that native grasses such as *L. hexandra* or introduced species such as the control grass (*U. arrecta*) constitute a useful forage resource to supply the dietary protein in grazing animals, one of the most limiting nutrients in different livestock scenarios under lowland tropics conditions.

Most of the studied variables were influenced by the measurement time. The gas production increased through the time. Additionally, TVFA, acetic, propionic and butyric acids were higher at 48 than 24 h. This is a recurrent result found in temperate and tropical grasses [51,54,55] since in the gas production technique, substrates are continuously being degraded by rumen microorganisms during the established incubation time. As the substrates degrade, a greater gas production and other fermentative products such as VFAs are expected to be generated [46].

Figure 2 shows a parallel comparison of the estimated chemical composition variables and the fermentation parameters of the studied grasses. An observed trend suggests that grasses with the highest CP (*U. arrecta* and *L. hexandra*) promote greater DMD, ammonia and TVFA. This is an expected behavior since dietary protein is rapidly degraded by ruminal microorganisms, increasing the ammonia concentration in the medium, and favoring the microbial cells synthesis and growth, which promote the substrate degradation efficiency and the TVFA production [46,56].

Figure 2 also shows that plants presented a similar NDF content; however the ADF and lignin showed a lower trend in the control grass. The slightly superior ADF content in native grasses would suggest a reduced DMD, TVFA concentration and a greater methanogenesis than the control grass [57,58]. However, this did not occur in the case of native grasses such as *L. hexandra* and *A. zizanioides*. This result suggests that possibly the indigestible component of the fiber fraction varied among the native grasses and that further studies must be performed to identify the real NDF digestibility and nutrient availability.

In general, the fermentation parameters of the experimental pastures were comparable with other improved grasses commonly used under different livestock systems in tropical grazing conditions. Native grasses DMD was similar to *Urochloa hybrid* cv. Cayman, *Mega-thyrsus maximus* cv. Mombaza, *Urochloa brizantha* cv. Toledo and *Urochloa decumbens* grasses, harvested between 42 and 65 days and fermented at 24 and 48 h [44,45]. Total gas production was higher than the values reported in *Megathyrsus maximus* (Jacq.) B.K.Simon and S.W.L.Jacobs, *Urochloa brizantha* (A. Rich.) R.D. Webster, *Urochloa decumbens* (Stapf) R.D. Webster, *Andropodon gayanus* Kunth, *Urochloa* riziziensis (R.Germ. and C.M. Evrard) Crins and *Pennisetum purpureum* Schumach grasses fermented during 24 h [59,60]. TVFA was higher in native plants at 48 h of fermentation compared with grasses such as *Andropogon gayanus* Kunth, *Urochloa decumbens* (Stapf) R.D. Webster, *Urochloa mutica* (Forssk.) T.Q.Nguyen, *Digitaria milanjiana* (Rendle) Stapf, *Megathyrsus maximus* (Jacq.) B.K.Simon and S.W.L.Jacobs, *Pennisetum purpureum* Schumach and *Setaria sphacelata* var. splendida harvested between 35 and 39 days of age [61]. The acetic, propionic and butyric acids were between the range observed in tropical grasses [44,61]; similarly, the observed methane and ammonia concentrations [50,62]. These results demonstrate the productive potential of the evaluated pastures, which constitutes them as a valuable forage resource to be considered as a food source for grazing cattle, during the flooding times occurring within the floodable savannah ecosystem in the Colombian Orinoquia.

## 5. Conclusions

In the “low” physiographic position of the Orinoquia floodplain savannahs, the studied grasses performed similar to other tropical forages in terms of biomass production or chemical composition. Remarkable native plants such as *L. hexandra* and *A. zizanioides* presented higher biomass production, CP, ash, cellulose and Ca levels than the control plant *U. arrecta*, an introduced species commonly used for grazing animals given its ability to withstand waterlogging conditions. The evaluated grasses can be harvested from 30 to 50 days of age to feed grazing animals without significant changes in nutrient availability.

In terms of fermentation characteristics, the studied grasses were also within the ranges reported in tropical pastures. The energy production efficiency was improved by the *L. hexandra* native grass due to the increase in TVFA and butyric acid concentrations. The latter effect was also observed in *A. zizanioides* grass. Likewise, the highest ammonia concentration was obtained with the *L. hexandra* native plant, with optimal values to ensure microbial cell synthesis on low digestibility tropical forages.

Native grasses such as *L. hexandra* and *A. zizanioides* constitute a valuable alternative forage resource during the flooding times of the studied ecosystem. Further research to evaluate fertilization responses and animal acceptability and productivity is still required.

## Figures and Tables

**Figure 1 animals-13-02760-f001:**
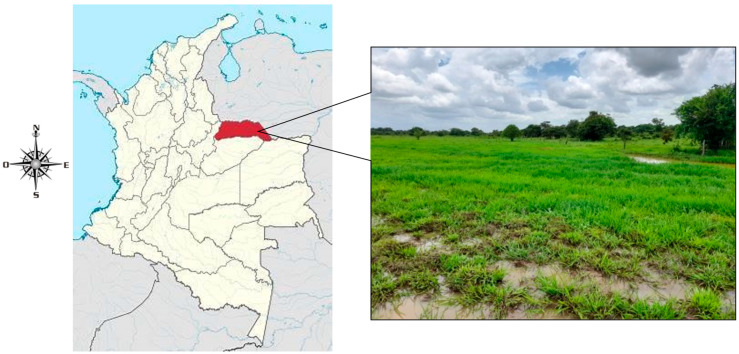
Arauca department in eastern Colombia (red color). Photographic: savannah floodplain ecosystem located in Arauca department.

**Figure 2 animals-13-02760-f002:**
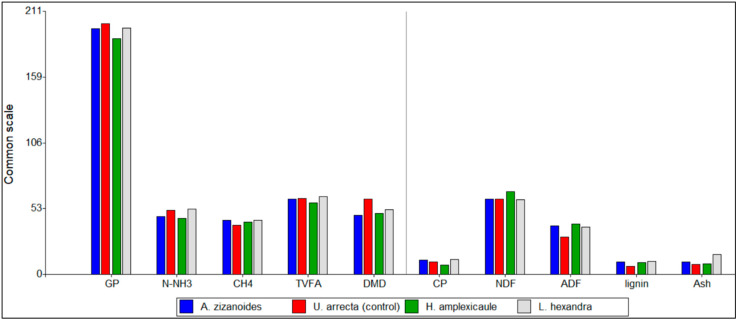
Comparison between chemical composition and fermentation parameters of the studied grasses at 48 h of incubation. GP: gas production (mL), N-NH3: ammonia (mg/dL), CH4: methane (mL/g DMD), TVFA: total volatile fatty acids (mmol/L), DMD: dry matter degradation (%), CP: crude protein (%), NDF: neutral detergent fiber (%), ADF; acid detergent fiber (%), lignin (%), Ash (%).

**Table 1 animals-13-02760-t001:** Botanic characteristics of the studied plants.

Species	Common Name	Growth Habit	Characteristic
*Leersia hexandra* Sw. (1788)	Lambedora grass	Rhizomatous-Stoloniferous	Native
*Acroceras zizanioides* (Kunth) Dandy (1931)	Black water straw	Stoloniferous	Native
*Hymenachne amplexicaulis* (Rudge) Nees (1829)	Water straw	Stoloniferous	Native
*Urochloa arrecta* (Hack ex T. Duran and Schinz) Stent (*control*)	Tanner grass	Stoloniferous	Introduced

**Table 2 animals-13-02760-t002:** Mean values and standard error of nutritional variables during the growth of grasses adapted to the lowland physiographic unit in the floodplain savannahs of the Colombian Orinoquia.

**Species**	**Age (days)**	**Height (cm)**	**GFY (Ton/ha)**	**DMY (Ton/ha)**	**DM (%)**	**CP (%)**	**NDF (%)**	**ADF (%)**
A. z	30	29.1 f ± 1.3	4.0 cd ± 0.31	1.1 de ± 0.1	27.8 c ± 1.0	11.6 ab ± 0.6	60.6 cd ± 1.1	38.8 ab ± 1.2
Control		47.8 de ± 2.3	3.9 cd ± 0.31	0.8 fg ± 0.1	19.8 f ± 1.0	10.1 bc ± 0.6	60.3 cd ± 1.1	29.8 d ± 1.2
H. a		68.9 b ± 2.6	3.6 d ± 0.31	0.7 g ± 0.1	18.8 f ± 1.7	7.5 cde ± 1.1	66.3 ab ± 1.8	40.6 a ± 2
L. h		45.0 de ± 1.9	4.4 cd ± 0.31	1.7 b ± 0.1	39.1 a ± 1.0	12.1 a ± 0.6	59.9 cd ± 1.1	37.8 abc ± 1.2
A. z	40	43.1 e ± 1.3	5.7 a ± 0.31	1.4 c ± 0.1	25.1 cde ± 1.0	11.3 ab ± 0.6	61.6 c ± 1.1	34.9 c ± 1.2
Control		64.6 bc ± 1.9	4.6 bc ± 0.31	1.1 e ± 0.1	23.0 e ± 1.0	8.2 cd ± 0.6	66.1 b ± 1.1	34.7 c ± 1.2
H. a		85.8 a ± 2.1	4.5 cd ± 0.31	1.0 ef ± 0.1	22.5 ef ± 1.0	7.2 de ± 0.6	67.7 ab ± 1.1	37.1 abc 1.2
L. h		59.9 c ± 1.5	5.7 a ± 0.31	1.9 ab ± 0.1	33.5 b ± 1.0	8.1 cd ± 0.6	57.5 de ± 1.1	35.4 bc ± 1.2
A. z	50	49.1 d ± 1.3	5.8 a ± 0.31	1.4 c ± 0.1	24.5 de ± 1.0	12.2 a ± 0.6	61.4 c ± 1.1	39.8 a ± 1.2
Control		67.7 b ± 1.9	4.8 bc ± 0.31	1.2 cde ± 0.1	26.2 cd ± 1.0	6.1 e ± 0.6	66.9 ab ± 1.1	34.6 c ± 1.2
H. a		89.0 a ± 2.1	5.4 ab ± 0.31	1.3 cd ± 0.1	24.5 de ± 1.0	6.6 de ± 0.6	69.6 a ± 1.1	39.1 a ± 1.2
L. h		63.5 bc ± 1.5	5.7 a ± 0.31	2.0 a ± 0.1	35.6 b ± 1.0	10.6 ab ± 0.6	56.6 e ± 1.1	37.2 abc ± 1.2
Interaction (*p*-value)		NS	NS	NS	0.0001	0.0066	0.0046	0.0199
**Species**	**Age (days)**	**Cellulose (%)**	**Hemicelulose (%)**	**Lignin (%)**	**Ash (%)**	**Ca (%)**	**P (%)**	
A. z	30	28.89 abc ±1.14	21.77 cd ± 0.84	9.9 ab ± 0.4	9.8 bc ± 0.9	0.54 a ± 0.05	0.26 bc ± 0.01	
Control		23.26 e ± 1.14	30.52 a ± 0.84	6.5 e ± 0.4	7.8 cd ± 0.9	0.46 abc ± 0.05	0.23 cd ± 0.01	
H. a		31.14 ab ± 1.97	25.65 b ± 1.46	9.5 abcd ± 0.8	8.3 bcd ± 1.2	0.23 c ± 0.1	0.28 abc ± 0.02	
L. h		27.42 bcd ± 1.14	22.04 c ± 0.84	10.4 a ± 0.4	15.8 a ± 0.9	0.44 abc ± 0.05	0.17 e ± 0.01	
A. z	40	25.87 cde ± 1.14	26.73 b ± 0.84	9.1 bcd ± 0.4	9.2 bc ± 0.9	0.59 a ± 0.05	0.28 ab ± 0.01	
Control		25.93 cde ± 1.14	31.43 a ± 0.84	8.8 bcd ± 0.4	7.2 cd ± 0.9	0.34 bc ± 0.05	0.23 cd ± 0.01	
H. a		28.05 abcd ± 1.14	30.64 a ± 0.84	9.0 bcd ± 0.4	6.7 d ± 0.9	0.26 c ± 0.05	0.27 bc ± 0.01	
L. h		27.35 bcd ± 1.14	22.15 c ± 0.84	8.1 d ± 0.4	13.8 a ± 0.9	0.35 bc ± 0.05	0.16 e ± 0.01	
A. z	50	31.37 a ± 1.14	21.59 cd ± 0.84	8.5 cd ± 0.4	10.9 b ± 0.9	0.49 ab ± 0.05	0.32 a ± 0.01	
Control		25.37 de ± 1.14	32.29 a ± 0.84	9.2 abcd ± 0.4	5.8 d ± 0.9	0.23 c ± 0.05	0.21 d ± 0.01	
H. a		29.5 ab ± 1.14	30.52 a ± 0.84	9.6 abc ± 0.4	6.5 d ± 0.9	0.28 c ± 0.05	0.25 bc ± 0.01	
L. h		27.72 bcd ± 1.14	19.34 d ± 0.84	9.5 abcd ± 0.4	14.6 a ± 0.9	0.28 c ± 0.05	0.14 e ± 0.01	
Interaction (*p*-value)		NS	<0.0001	0.0004	0.0476	NS	0.0244	

A. z: *Acroceras zizanioides* (Kunth) Dandy (1931); Control: *Urochloa arrecta* (Hack ex. T. Duran & Schinz); H. a: *Hymenachne amplexicaulis* (Rudge) Nees (1829); L. h: *Leersia hexandra* (Sw. (1788); GFY: green forage yield; DMY: dry matter yield; CP: crude protein; NDF: neutral detergent fiber; ADF: acid detergent fiber; DMD: dry matter digestibility; Ca: calcium; P: phosphorus. S.E: standard error. Different letters in the same column differ statistically (*p* < 0.05). NS: not significant.

**Table 3 animals-13-02760-t003:** Mean values and standard error of gas volume and dry matter degradation of the studied grasses at 12, 24 and 48 h of fermentation.

Species	Time	Gas Production (mL/g DM)	DMD (%)	pH
*A. zizanioides*	12	126.7 ± 2.75	25.5 f ± 1.2	6.7 ± 0.02
*U. arrecta (control)*		124.7 ± 2.75	31.9 e ± 1.2	6.8 ± 0.02
*H. amplexicaulis*		120.7 ± 2.75	20.8 g ± 1.2	6.8 ± 0.02
*L. hexandra*		125.3 ± 2.75	26.8 f ± 1.2	6.7 ± 0.02
*A. zizanioides*	24	154.3 ± 2.75	37.1 d ± 1.2	6.6 ± 0.02
*U. arrecta (control)*		160.0 ± 2.75	48.8 bc ± 1.2	6.7 ± 0.02
*H. amplexicaulis*		147.7 ± 2.75	35.4 de ± 1.2	6.6 ± 0.02
*L. hexandra*		158.0 ± 2.75	38.9 d ± 1.2	6.6 ± 0.02
*A. zizanioides*	48	197.3 ± 2.75	47.5 c ± 1.2	6.6 ± 0.02
*U. arrecta (control)*		201.7 ± 2.75	60.6 a ± 1.2	6.6 ± 0.02
*H. amplexicaulis*		189.7 ± 2.75	49.1 bc ± 1.2	6.6 ± 0.02
*L. hexandra*		198.0 ± 2.75	51.8 b ± 1.2	6.6 ± 0.02
*p*-value	Specie	0.0260	0.0001	NS
	Time	<0.0001	<0.0001	<0.0001
	Interaction	NS	0.0178	NS

Different letters in the same column indicate significant differences (*p* < 0.05). NS: not significant.

**Table 4 animals-13-02760-t004:** Mean values and standard error of ammonia, methane and volatile fatty acids of the evaluated grasses at 24 and 48 h of fermentation.

Species	Time	Ammonia (mg/dL)	Ch_4_ (ml/gDMD)	TVFA (mmol/L)	Acetic (mmol/L)	Propionic (mmol/L)	Butyric (mmol/L)
*A. zizanioides*		38.5 d ± 1.1	31.5 bc ± 1.7	26.1 ± 1.7	15.5 ± 1.0	8.4 ± 0.6	2.2 ± 0.2
*U. arrecta (control)*		42.8 c ± 1.2	29.6 bc ± 1.7	24.6 ± 1.7	14.0 ± 1.0	8.7 ± 0.6	1.8 ± 0.2
*H. amplexicaulis*	24	38.5 d ± 1.1	33.8 b ± 1.7	21.5 ± 1.7	13.0 ± 1.0	6.7 ± 0.6	1.8 ± 0.2
*L. hexandra*		42.2 c ± 1.1	27.8 c ± 1.7	28.7 ± 1.7	17.6 ± 1.0	8.6 ± 0.6	2.4 ± 0.2
*A. zizanioides*		46.6 b ± 1.1	43.5 a ± 1.7	60.4 ± 1.7	36.3 ± 1.0	16.9 ± 0.6	7.1 ± 0.2
*U. arrecta (control)*		51.6 a ± 1.2	39.6 a ± 1.7	60.8 ± 1.7	36.2 ± 1.0	17.6 ± 0.6	7.0 ± 0.2
*H. amplexicaulis*	48	45.1 bc ± 1.1	41.8 a ± 1.9	57.7 ± 1.7	35.8 ± 1.0	15.8 ± 0.6	6.7 ± 0.2
*L. hexandra*		52.6 a ± 1.1	43.5 a ± 1.7	62.3 ± 1.7	37.8 ± 1.0	16.9 ± 0.6	7.6 ± 0.2
	Species	0.011	NS	0.029	NS	NS	0.0069
*p*-value	Time	<0.0001	<0.0001	<0.0001	<0.0001	<0.0001	<0.0001
	Interaction	<0.0001	0.0437	NS	NS	NS	NS

DMD: dry matter digestibility; S.E: standard error; TVFA: total volatile fatty acids. Different letters in the same column indicate significant differences (*p* < 0.05). NS: not significant.

## Data Availability

The data are available upon reasonable request to the first author.

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
