# Peer review of "Chemical Composition and In Vitro Ruminal Fermentation Characteristics of Native Grasses from the Floodplain Lowlands Ecosystem in the Colombian Orinoquia"

_animals, 2023, doi:10.3390/ani13172760_

Round 1

Reviewer 1 Report

If the grass samples were collected at each cutting time using a 1 m2 PVC frame on each sampling date, a grass sample was collected from each experimental plot, How was the place to take the sample decided? The frame was launched to have a random sample or a specific place was decided to take the sample within the plot maybe to the center of the plot?

The research provides practical information but I consider that it should be complemented with some other determinations such as the calculation of the degradability of the OM, CP, NDF, ADF and the digestible or metabolizable energy of each grass species per hectare and at each growing time, besides the DMD. The information could also be complemented by determining through the gas technique to determine the digestion rates and soluble, potentially digestible, degradable and non-degradable fractions in the rumen (Orskov and McDonals).

In general, research to characterize native grasses provides important information, but I think it falls short for a scientific article, although it may be publishable as a short communication.

Author Response

First reviewer’s responses

Dear reviewer

The authors appreciate the insightful comments.

We attach all corrections and answers.

Comment

If the grass samples were collected at each cutting time using a 1 m2 PVC frame on each sampling date, a grass sample was collected from each experimental plot, How was the place to take the sample decided? The frame was launched to have a random sample or a specific place was decided to take the sample within the plot maybe to the center of the plot?

Response

Three sampling points within each plot were randomly selected, and the cut-off ages (30, 40 and 50 days) were assigned to each measurement point randomized. These points were used to take grasses samples, using a 1 m2 PVC frame  by cutting the available material within the frame at 10 cm from the ground.

A better description was made in the text.

The research provides practical information but I consider that it should be complemented with some other determinations such as the calculation of the degradability of the OM, CP, NDF, ADF and the digestible or metabolizable energy of each grass species per hectare and at each growing time, besides the DMD. The information could also be complemented by determining through the gas technique to determine the digestion rates and soluble, potentially digestible, degradable and non-degradable fractions in the rumen (Orskov and McDonals).

Response

Thank you for your comment. We emphasize that this study is a first approximation to understand the nutritional composition and nutrient supply of these native grasses. The measurements carried out were made with the NIRS method and for the variables calibrated in the equipment. Unfortunately, the variables that are suggested to be included were not available.

The other option would be to estimate the components degradability (PC; NDF; FDA, etc) from the value of the DMD, however we consider that it is not appropriate. The DMD is an indicator of the digestibility of all the components in general (where nutrients interactions exist) and not individually. Hence, the literature reports studies focused only on defining, for example, the NDF digestibility or the protein degradability according to its fractions (soluble, B and C), among others. Thus, we consider that in order to evaluate the degradability of the different components individually, it would be necessary to carry out specific tests that were not done in this study.

On the other hand, in the gas production technique, models were not adjusted to determine digestion rates or degradable or non-degradable fractions because the fermentation process only lasted 48 hours. If the average values of gas production per treatment are observed up to the 48 hours, only a linear increasing behaviour is observed, without reaching the asymptote of the fermentation process. In this way, although we are aware that the suggested parameters can be estimated with 48 h, we consider that a longer incubation time (72 or 96 h) would be required for a more accurate estimate.

Thanks again for your comments and we will take them into account in future studies.

In general, research to characterize native grasses provides important information, but I think it falls short for a scientific article, although it may be publishable as a short communication.

Response

OK

Reviewer 2 Report

Dear authors

I have reviewed your manuscript. In this regard I would like to make some comments. It sounds interesting to evaluate the grasses that grow in flood zones. For this, in vitro fermentation studies allow to have a vision of its effects at the rumen level. However, I find your research lacking in depth. I would highly recommend including changes in the rumen microbiota, as well as reporting the metabolizable and kinetic energy of gas production, for example. Much of the discussion focuses on comparing, rather than explaining the observed phenomenon. I highly recommend deeper discussion. The paragraphs from lines 78 to 96 of the introduction should be deleted.

The tables need to be improved. Figure 2 is not necessary.

Regards

English language needs to be improved.

Author Response

Second reviewer’s responses

Dear reviewer

The authors appreciate the insightful comments.

We attach all corrections and answers.

Comment

Dear authors

I have reviewed your manuscript. In this regard I would like to make some comments. It sounds interesting to evaluate the grasses that grow in flood zones. For this, in vitro fermentation studies allow to have a vision of its effects at the rumen level. However, I find your research lacking in depth.

I would highly recommend including changes in the rumen microbiota, as well as reporting the metabolizable and kinetic energy of gas production, for example.

Response: models were not adjusted to determine digestion rates or degradable or non-degradable fractions because the fermentation process only lasted 48 hours. If the average values of gas production per treatment are observed up to the 48 hours, only a linear increasing behavior is observed, without reaching the asymptote of the fermentation process. In this way, although we are aware that the suggested parameters can be estimated with 48 h, we consider that a longer incubation time (72 or 96 h) would be required for a more accurate estimate.  Thanks for your comments and we will take them into account in future studies.

Much of the discussion focuses on comparing, rather than explaining the observed phenomenon. I highly recommend deeper discussion.

Response:

The discussion was based on the obtained results and emphasis was made to compare the native grasses parameters against improved species, to see its usefulness in animal nutrition.

Similarly, we revised the section and tried to complement it

The paragraphs from lines 78 to 96 of the introduction should be deleted.

Response: this section was summarized trying to accommodate all the suggestions received

The tables need to be improved. Figure 2 is not necessary.

Response: The tables were modified for greater clarity. The figure 2 was deleted.

The changes were incorporate in the text

English language needs to be improved.

Response:

Corrected

Reviewer 3 Report

Dear Authors,

    Thanks for sharing your work to the readers. I would like to suggest you some advice before publication, please it as follows:

    (1) When evaluating the in vitro fermentation characteristics, at least the OM digestibility or DM digestibility should be provided. Moreover, general VFA includes acetate, propionate, isobutyrate, butyrate, isovalerate, valerate. Please add these icformation to have a better explanation for the results.

  (2) For Line 88-96, it is not suitable to allow so many word for the technology of in vitro incubation, as it is common for readers in this field.  Instead, I would like suggest you with more updated reference in this topic.

  (3) Some minor mistakes should be cautious, for instance, Line 184, 

Variables like dry matter degradation (DMD), gas volume and pH were evaluated al, here, al should be "at".

Minor editing of English language required, for instance, the spell.

Author Response

Third reviewer’s responses

Dear reviewer

The authors appreciate the insightful comments.

We attach all corrections and answers.

Comment

Dear Authors,

Thanks for sharing your work to the readers. I would like to suggest you some advice before publication, please it as follows:

  • When evaluating the in vitro fermentation characteristics, at least the OM digestibility or DM digestibility should be provided. Moreover, general VFA includes acetate, propionate, isobutyrate, butyrate, isovalerate, valerate. Please add these icformation to have a better explanation for the results.

Thank you for your comment. In the second part of in vitro fermentation the dry matter digestibility is provided in each sampling time.

Unfortunately, in this first approximation we could only evaluate the main 3 AGV (acetate, butyrate, and propionate). However, we will take your comment into account in future studies.

  • For Line 88-96, it is not suitable to allow so many word for the technology of in vitro incubation, as it is common for readers in this field.  Instead, I would like suggest you with more updated reference in this topic.

Response: Corrected

  (3) Some minor mistakes should be cautious, for instance, Line 184, 

Variables like dry matter degradation (DMD), gas volume and pH were evaluated al, here, al should be "at".

Response: Corrected

Minor editing of English language required, for instance, the spell.

Response: Corrected

Reviewer 4 Report

The authors' work addresses an important issue in cattle breeding. In my opinion, the work needs refinement. 

Please check the journal's guidelines for published papers and the size of the abstract, for example. In my opinion, the purpose of the work presented and the most relevant results should be presented in the abstract. 

The literature review in my opinion is very cursory and poor. I would suggest enriching this chapter with information on what chemical components are required for cattle. What is the optimal diet for cows and how does it translate into weight gain (including fat and muscle mass), examples of chemical compositions of grasses (from different regions of the world along with highlighting the importance of grazing cattle for food production in the world and the global market share). 

Tables 3 and 4 were not discussed in the paper. 

The graph (Figure 2) lacks a standard deviation. Also in the tables presented by the authors. I miss the content of cellulose and hemicelluloses in the brad material. 

Author Response

Fourth reviewer’s responses

Dear reviewer

The authors appreciate the insightful comments.

We attach all corrections and answers.

Comment

The authors' work addresses an important issue in cattle breeding. In my opinion, the work needs refinement. 

Please check the journal's guidelines for published papers and the size of the abstract, for example. In my opinion, the purpose of the work presented and the most relevant results should be presented in the abstract. 

The literature review in my opinion is very cursory and poor. I would suggest enriching this chapter with information on what chemical components are required for cattle. What is the optimal diet for cows and how does it translate into weight gain (including fat and muscle mass), examples of chemical compositions of grasses (from different regions of the world along with highlighting the importance of grazing cattle for food production in the world and the global market share). 

Response:

The abstract was written showing the most important methods and results of the study.

The introduction was improved considering some of your suggestions. We consider that mention the importance of grazing is a well-documented topic and it seems more pertinent to highlight its importance in the studied region.  The same ocurred with the chemical composition of the forages. In the discussion, a more detailed comparison with other forages is made, so adding this information in the introduction would make the document repetitive.

Tables 3 and 4 were not discussed in the paper. 

The graph (Figure 2) lacks a standard deviation. Also in the tables presented by the authors. I miss the content of cellulose and hemicelluloses in the brad material. 

Response: Tables 3 and 4 were referenced in the following paragraph:

“Finally, DMD, ammonia and methane variables varied by the species x measurement time interaction. Fermentation parameters of the interaction are shown in tables 3 and 4. U. arrecta (control) presented a superior DMD compared with the native species in all of the evaluated measurement times (p < 0.05), with values of 31.9, 48.8 and 60.6 % at 12, 24 and 48 h, respectively. Native grasses showed DMD mean values of 24.4, 37.1 and 49.5 % at 12, 24 and 48 h, respectively. The ammonia was higher at 24 and 48 h in U. arrecta (control) and L. hexandra species (p < 0.05) than the other grasses, with values ranging from 42.2 to 52.6 mg/dL. Methane at 24 and 48 h of fermentation was similar between the estu-died plants. The only difference observed was at 12 h of fermentation where H. amplexicaulis presented higher methane concentrations than L. hexandra species (p < 0.05; 33.8 vs 27.8 ml/g DMD).”

Figure 2 was eliminated according to the suggestions of one of the evaluators.

The cellulose and hemicelluloses were included and discussed.

Round 2

Reviewer 1 Report

No more comments

Reviewer 3 Report

The authors have addressed my concerns in an acceptable way. I think it is ready for publication. Good luck!